# Optimal Cash Holding Ratio for Non-Financial Firms in Vietnam Stock Exchange Market

**Cuong Nguyen Thanh**

Faculty of Accounting and Finance, Nha Trang University, Nha Trang City 650000, Vietnam;
Cuongnt@ntu.edu.vn

**Abstract:** The purpose of this research is to investigate whether there is an optimal cash holding ratio, in which firm's performance can be maximized. The threshold regression model is applied to test the threshold effect of the cash holding ratio on firm's performance of 306 non-financial companies listed on the Vietnam stock exchange market during the period of 2008–2017. Experimental results showed that a single-threshold effect exists between the ratio of cash holding and company's performance. A proportion of cash holding within a threshold of 9.93% can contribute to improvement of the company's efficiency. The coefficient is positive but tends to decrease when the cash holding ratio passes the 9.93% check point, implying that an increase in cash holdings ratio will continue to diminishment of efficiency eventually. Therefore, the relationship between cash holding ratio and firm's performance is nonlinear. From this result, this paper provides policy implications for non-financial companies listed on the Vietnam stock exchange market in determining the proportion of cash holding flexibly. In detail, non-financial companies listed on the Vietnam stock exchange market should not keep the cash holding ratio over 9.93%. To ensure and enhance the company's performance, the optimal range of cash holding ratios should be below 9.93%.

**Keywords:** cash holding ratio; firm's efficiency; threshold regression model; non-financial companies; Vietnam stock exchange market

## 1. Introduction

The amount of cash held plays an important role in most companies because it provides the ability to pay in cash and directly affects the performance of the company. The amount of cash held by the company is understood as cash and cash equivalents such as bank deposits and short-term securities which are able to be quickly converted into money (Bates et al. 2009; Ferreira and Vilela 2004). If the company holds a large amount of cash, the opportunity cost will arise. However, if a company holds too little cash, it may not be enough to cover the regular expenses. Therefore, the amount of cash held by the company must be sufficient to ensure regular operations solvency, contingency for emergencies, and also future projections (if needed). Dittmar and Mahrt-Smith (2007) stated that in 2003, the sum of all cash and cash equivalents represented more than 13% of the sum of all assets for large US firms. Al-Najjar and Belghitar (2011) found that cash represents, on average, 9% of the total assets for UK firms. In Vietnam context, the cash and cash equivalents account for more than 10% of total assets of firms. Thus, cash represents a sizeable asset for firms. Cash management may therefore be a key issue for corporate financial policies.

Regarding this topic, the first studies looked at antecedents of corporate cash holdings (Ferreira and Vilela 2004; Kim et al. 1998; Opler et al. 1999; Ozkan and Ozkan 2004). Most of these papers assume that cash holding is determined by the firm characteristics (i.e., leverage, growth opportunities, cash flow, investment in fixed assets, and size of the firm) and industry sector. In addition, Sheikh and Khan (2016) showed that cash holding is determined not only by the firm characteristics but

also by the manager characteristics (i.e., age, gender, whether or not the manager is a chief executive officer, and whether or not the manager is a board director). Besides, corporate governance may also affect the value of a firm's cash holdings. Evidence by Dittmar and Mahrt-Smith (2007) shows that investors value the excess cash holdings of well-governed firms at nearly double the value of the poorly governed firms.

Despite the increasing amount of literature on corporate cash holding, there are not many studies focusing on the relationship between cash holding and company performance. At the same time, corporate cash holdings have benefits and costs for the firm and, consequently, an optimum cash level may exist at which the performance of the firm is maximized. Evidence by Martínez-Sola et al. (2013) shows that in US industrial firms there exists an optimal cash holding ratio. This finding is also consistent with Azmat (2014), which found the existence of optimal cash level for a sample of listed Pakistani firms. Following this optimal level, firms will adjust their cash reserve to maximize firm value. On the other hand, most of the research work has been carried out in developed economies and very little is known about the cash holding of firms in developing economies.

According to Horioka and Terada-Hagiwara (2014), Asian firms are heavily constrained by borrowing limits and will hold more cash for future investments than firms in developed countries. Hence, our focus on an emerging country, Vietnam, allows us to offer a number of new insights beyond the existing studies of the relationship between cash holdings and firm's performance. In Vietnam context, given the great opportunities and challenges now, companies need to focus on cash management, as lifeblood of the company. Therefore, the current question for listed companies on the Vietnam stock exchange market is how to manage cash in order to improve operational efficiency and contribute to increase of the company's value. To solve this problem, listed companies on the Vietnam stock exchange market need to know how the cash holding ratio affects the firm's performance.

In this study, the goal is to indicate to what extent the cash holding ratio will have a positive effect on increase of the company's performance and to what extent the cash holding ratio will have a negative effect on reduction of company's performance. Different from previous studies, this study applies the threshold regression model of Hansen (1999) to build the model to investigate the impact of cash holding ratio on the performance of the listed companies on the Vietnam stock exchange market. The results show empirically that an optimum level of cash holdings exists where firm's performance is maximum, for a sample of 306 listed non-financial Vietnamese companies during 2008–2017. Deviations from the optimum level reduce firm value. It means that firms should balance the costs and benefits of cash holdings to find the optimal cash level to maximize firm's performance and value. If a firm has a cash holding above or below the optimal level, its performance and value will decrease. The results of this study can help managers of listed companies on the Vietnam stock exchange market to adjust the cash holding ratio to improve operational efficiency and contribute to the increase of the company's value.

To the best of author's knowledge, until now, there has been no published research on the application of threshold regression model to study this relationship. Hence, by using the threshold regression model developed by Hansen (1999), this study further fills the gap in the literature on the behavior of firms, and focuses on evidence of cash policies and firm's performance.

The paper includes five parts: Section 1 introduces research issues; Section 2 presents a theoretical overview and a research model; Section 3 presents data collection and methods; Section 4 presents the results of empirical research; the final section summarizes the findings and implications for cash management.

## 2. Literature Reviews and Hypothesis

Making a decision to hold an amount of cash in the company will affect efficiency or dynamics and corporate value. Each company has different reasons for holding cash; according to previous studies, there are major motives for the company to hold cash, that is:

Trading motive: Cash is the mean of exchange, so companies need cash to conduct daily transactions; however, the demands for cash of different companies are not the same (Bates et al. 2009; Opler et al. 1999). According to Nguyen et al. (2016), when firms have insufficient internal funds or liquid assets, they will raise funds from external capital markets, liquidate existing assets, limit dividend payouts, and reduce investment opportunities. However, all of these activities are costly.

Prevention motive: In addition to keeping cash for daily transactions, companies also need cash reserves for unexpected spending needs. According to Ferreira and Vilela (2004), the company holds cash to finance financial or investment activities when other resources are unavailable or very costly; however, each company has various demands for cash reserves in obtaining different objectives depending on situations.

Signal motive: Because of information asymmetry between managers and shareholders, managers will signal the prospects of the company to investors through their dividend payment policy. According to Harford (1999), the payment of dividends is more positive than the purchase of treasury shares by cash, because dividend payment generates a signal of a commitment to a higher dividend payment in the future; meanwhile, buying treasury securities is considered to be this year's event only, and should not continue in the next year.

Represent motive: Managers can decide whether the company will withhold cash or pay dividends to shareholders. The free cash flow might increase discretion by managers, which goes against shareholders' interest (Jensen 1986). The study of Harford (1999) confirmed that companies with large amounts of cash tend to spend a lot of money to conduct numerous acquisitions. Furthermore, the study of Blanchard et al. (1994) provided evidence that firms do not pay dividends during a period of time but making zero acquisitions will spend their cash in many other investment activities.

The corporate cash holding determinants have been a subject of explanation in the framework of three theories, namely the trade-off model, pecking order theory, and free cash flow theory.

The free cash flow theory suggests that managers hold cash to serve their own interests, thus increasing the conflict between investors and company's managers (Harford 1999; Jensen 1986). The theory of free cash flow also highlights the representative cost of holding cash. Companies with high growth opportunities have high agency costs, so they will tend to store more cash in order to be proactive in their capital. If there is a conflict between management and shareholders, management tends to store as much cash as possible to pursue their goals. Cash can be paid not only for making profits, but also for projects where investors are not ready to raise capital. Moreover, the board can also hold cash because of risk aversion.

The pecking order theory of Myers and Majluf (1984) suggests that managers can decide the order of capital financing to minimize the cost of information asymmetry and other financial costs. This theory implies that companies prefer internal financing. The directors adjust the dividend payout ratio to avoid the sale of ordinary shares, preventing a major change in the number of shares. If external funding is available, Myers and Majluf (1984) believed that the safest securities should be issued first. Specifically, debt is usually the first security to be issued and equity sold outside is the last solution. However, Myers and Majluf (1984) also argued that there will be no optimal level of cash holdings, but holding cash should serve as a buffer between retaining profits and investment needs.

The trade-off theory suggests that companies can finance by borrowing or retaining cash and they all have their advantages and costs. With the trade-off theory, also called transaction cost model of Opler et al. (1999), the company can determine a level of cash holdings by balancing the marginal cost of holding highly liquid assets and the profit margins of holding cash. Profit margins of cash holdings will reduce the likelihood of financial distress, allowing the company to make optimal investments and avoiding the costs incurred by external funding or liquidation of assets of company. Because the market is imperfect, it is difficult for companies access the capital market or to bear the cost of external funding. The marginal cost of holding cash is the opportunity cost of holding cash when it offers less benefit than investing in an equal risk condition (Ferreira and Vilela 2004; Opler et al. 1999). When companies need cash to meet their expenses, they need external funding from the capital market

or liquidate their assets. Since the capital markets are imperfect, transaction costs can be avoided by holding an optimal cash level.

Based on the benefits and costs of holding cash, there have been a few recent studies on the relationship between the amount of cash held and the performance or value of the company. The real impact of cash on firm's performance or value is still being debated on the basis of empirical theory and evidence, creating many different perspectives.

The first view is that a high ratio of cash reserves will reduce the performance or value of the company. Supporting this point of view, Harford (1999) examined the relation between a firm's acquisition policy and its cash holding. It was shown by the results that firms with a large amount of cash are more likely to make acquisitions that will decrease operational efficiency and firm value. The results of this empirical study are explained based on the theory of free cash flow. It means that managers of firms with a large amount of cash desire to increase the scope of their authority. In another study, Harford et al. (2008) concluded that firms with poor governance will spend more cash than other similar firms since the entrenched managers will prefer to overinvest rather than reserving cash for firms. Therefore, firms with a high level of cash holdings will have a lower firm performance or value.

The second opinion supports the existence of a positive relationship between business value and the amount of cash held. Saddour (2006) studied the relationship between firm's value and the amount of cash held on the French stock exchange market in the period of 1998–2002. The results indicate that high cash reserves will increase operational efficiency or company's value. Similarly, Bates et al. (2009) also found evidence that firms hold more cash when firms' cash flow becomes riskier. This evidence strongly supports the precautionary motivations of cash holdings and implies a positive relationship between firm value and cash holdings.

The third view believes in a nonlinear relationship between cash holding and firm's performance or value. Supporting this perspective, Martínez-Sola et al. (2013) used US industry's data from 2001 to 2007, and found a nonlinear relationship between cash holding ratio and company's value. They explained that the concave relationship between cash holdings and firm value exists because firms balance the costs and benefits of cash holdings to identify the optimal level of cash. Following this optimal level, firms will adjust their cash reserve to maximize the firm value. This result was also discovered earlier by Azmat (2014), who found the existence of optimal cash level for a sample of listed Pakistan firms from 2003 to 2008. In Vietnam, Nguyen et al. (2016) investigated the nonlinear relationship between firm value and corporate cash holdings in a sample of non-financial Vietnamese firms from 2008 to 2013. Authors focused on both static and dynamic regressions to test for a nonlinear relationship. Their results reveal an "inverse U-shape" relationship between firm value and cash holdings, which is in line with the trade-off theory.

According to the trade-off theory, in the context of Vietnam firms, author still expect there is a nonlinear relationship between cash holdings and company's performance. Agreeing with thes studies above, for this relationship, we set the hypothesis as following:

**Hypothesis 1 (H1):** *There is a nonlinear relationship between cash holdings and company's performance in Vietnamese listed non-financial companies.*

## 3. Data and Methodology

### 3.1. Data and Sample Collection

Data includes the annually audited financial statements which can be collected from the website: https://vietstock.vn/. The companies selected for the sample are active non-financial companies, with full financial reporting for the period of 2008–2017. With this sampling method, data collected includes 306 non-financial companies operating in the 2008–2017 period. Consequently, the final dataset is a strongly balanced panel dataset, which includes 3060 firm-year observations of 306 companies (306 companies × 10 periods = 3060 observations).

*3.2. Variables*

3.2.1. Dependent Variable

When studying the relationship between firms' cash holdings and performance, to measure firms' performance, Shinada (2012) used the return on asset (denoted by ROA). Iftikhar (2017) and Vijayakumaran and Atchyuthan (2017) also used the ROA to measure company's performance. Agreeing with these above studies, we used the book value to calculate firms' performance. The measurement of firms' performance was defined as below:

$$\text{ROA} = \frac{\text{Profit before tax and interest}}{\text{Total assets}} \tag{1}$$

3.2.2. Threshold and Explanatory Variables

There are two categories of explanatory variables in our panel data and threshold regression model. One is the threshold variable, which is the key variable used to assess the optimal cash holding ratio of a firm and to capture the threshold effect of cash holding on company's performance.

The threshold variable is a variable. When the threshold variable is bigger or smaller than the threshold value ($\gamma$), the samples can be divided into two groups, which can be expressed in different slopes $\beta_1$ and $\beta_2$. The explanatory variable is a variable, reflecting its impact on the dependent variable. In the threshold regression model, explanatory variable impacts are not fixed but depend on the threshold value of the threshold variable.

In this study, the measurement of threshold and independent variables through the cash holdings ratio (denoted by CASH) was performed. Following previous studies (Azmat 2014; Martínez-Sola et al. 2013; Nguyen et al. 2016; Opler et al. 1999; Vijayakumaran and Atchyuthan 2017), the cash holdings ratio was calculated as cash and cash equivalents divided by total assets. We used the book value to calculate the cash holdings ratio. The measurement of firms' cash holdings ratio was defined as below:

$$\text{CASH} = \frac{\text{Cash and cash equivalents}}{\text{Total assets}} \tag{2}$$

3.2.3. Control Variables

On the basis of previous studies (Azmat 2014; Martínez-Sola et al. 2013; Nguyen et al. 2016; Vijayakumaran and Atchyuthan 2017), our threshold regression model includes several additional variables to control for a set of firm-specific characteristic that are likely to be correlated with company's performance. These include firm size (denoted by SIZE), leverage (denoted by LEV), and firm's growth (denoted by MB). The following section will analyze interconnection between these variables relative to company's performance.

Firm size (SIZE): According to Dang et al. (2018), in empirical corporate finance, the firm size is commonly used as an important, fundamental firm characteristic. They examined the influences of employing different proxies (total assets, total sales, and market capitalization) of firm size. The results show that, in most areas of corporate finance, the coefficients of firm size measures are robust in sign and statistical significance. In addition, the coefficients on repressors other than firm size often change sign and significance when different size measures are used. Therefore, the choice of size measures needs both theoretical and empirical justification.

The firm size is considered one determinant of firm performance and value. Abor (2005) and Vijayakumaran and Atchyuthan (2017) suggested that enterprises of higher size generally have higher firm performance. On the other hand, researches by Cheng et al. (2010), Martínez-Sola et al. (2013), and Nguyen et al. (2016) suggest that enterprises of higher size generally have lower firm performance and value. Thus, the relationship between the size and the performance of companies is unclear. To measure the firm size, there exist different perspectives. According to Azmat (2014), Nguyen et al. (2016), and Vijayakumaran and Atchyuthan (2017), the firm size is defined by a natural logarithm

of total assets. Further, Martínez-Sola et al. (2013) showed that the firm size is defined by natural logarithm gross sales. In this study, we only used the book value of total asset to calculate the firm size. The measurement of firm size was defined as below:

$$\text{SIZE} = \text{Ln(Book value of Total assets)} \tag{3}$$

Growth (MB) is considered to be a factor related to firm performance. Abor (2005) suggested that enterprises of higher growth opportunities generally have higher profitability. On the other hand, researches by Nguyen et al. (2016) suggest that enterprises of higher growth generally have lower firm performance and value. In addition, Vijayakumaran and Atchyuthan (2017) suggested that sales growth is not significantly related to firm performance. Thus, the relationship between the growth and the firm performance is unclear. To measure growth, there exist different perspectives. According to Abor (2005), Nguyen et al. (2016), and Vijayakumaran and Atchyuthan (2017), growth is defined by the growth rate on operating sales. Further, Cheng et al. (2010) showed that growth is defined by the growth rate of operating sales and growth rate of total assets. In this study, growth was measured by market value over the book value of stocks. The measurement of growth was defined as below:

$$\text{MB} = \frac{\text{Market value of stocks}}{\text{Book value of stocks}} \tag{4}$$

Leverage (LEV) is considered one determinant of firm performance and value. Abor (2005), Nguyen et al. (2016), and Vijayakumaran and Atchyuthan (2017) suggested that enterprises of higher leverage generally have lower profitability. On the other hand, researches by Martínez-Sola et al. (2013) suggest that enterprises of higher leverage generally have higher firm value. In addition, the empirical results by Cheng et al. (2010) strongly indicate that triple-threshold effect exists between leverage and firm value. Thus, the relationship between the leverage and the firm performance is unclear. To measure leverage, there exist different perspectives. According to Abor (2005), Azmat (2014), and Nguyen et al. (2016), leverage is defined by total debt over total assets. Further, Martínez-Sola et al. (2013) and Vijayakumaran and Atchyuthan (2017) showed that leverage is defined by total debt over total equity. In this study, we only used the book value of total debt and total asset to calculate leverage. The measurement of leverage was defined as below:

$$\text{LEV} = \frac{\text{Market value of total debt}}{\text{Book value of total assets}} \tag{5}$$

### 3.3. Models and Estimation Methods

This study aimed to test whether there is an optimal threshold between the cash holding ratio and company's performance. According to the trade-off theory, the optimal ratio of cash holdings is determined by a trade-off between marginal cost and profit margin of cash holdings (Opler et al. 1999). Therefore, this study assumed the existence of an optimal ratio of cash holdings, and tried to use the threshold regression model to estimate this ratio. To test the hypothesis, this study applied the threshold regression model of Hansen (1999). Single-threshold and multi-threshold models were based on the threshold regression model of Hansen (1999) as follows.

The single-threshold regression model was shown as:

$$\text{ROA}_{i,t} = \begin{cases} \mu_i + \theta' H_{i,t} + \beta_1 \text{CASH}_{i,t} + \varepsilon_{i,t} & \text{if CASH}_{i,t} \leq \gamma \\ \mu_i + \theta' H_{i,t} + \beta_2 \text{CASH}_{i,t} + \varepsilon_{i,t} & \text{if CASH}_{i,t} > \gamma \end{cases}, \tag{6}$$

where $\theta' = (\theta_1, \theta_2, \theta_3)'$ and $H_{i,t} = (\text{SIZE}_{i,t}, \text{MB}_{i,t}, \text{LEV}_{i,t})$; $\text{ROA}_{i,t}$ represents for firm's performance, measured by profit before tax and interest on total assets; $\text{CASH}_{i,t}$ represents the proportion of cash held by the company, measured by the ratio of cash and cash equivalents on total assets; $(\text{CASH}_{i,t})$ is the explanatory variable and also the threshold variable, estimated at each different threshold;

$H_{i,t}$ are control variables that affect company performance, including company size ($SIZE_{i,t}$), company growth ($MB_{i,t}$) and leverage ($LEV_{i,t}$); $\theta_1$, $\theta_2$, and $\theta_3$ are the estimated regression coefficients of the corresponding control variables; $\gamma$ is the value of the estimated threshold; $\mu_i$ is a fixed effect representing the heterogeneity of companies operating under different conditions; $\beta_1$ and $\beta_2$ are regression coefficients of the proportion of cash held by the company; the error $\varepsilon_{i,t}$ is assumed to be independent and has a normal distribution $\left(\varepsilon_{i,t} \sim \text{iid}(0, \sigma^2)\right)$; i represents different companies; t represents different periods.

According to Hansen (1999), for estimating procedures, this study first removed the fixed effect ($\mu_i$) by using the techniques of estimating "internal transformation" in a traditional fixed-effects model for panel data. By using the ordinary least square estimation method and minimizing the sum of squared error ($S_1(\gamma)$), the test can obtain the estimation of the threshold value ($\hat{\gamma}$) and the residual variance ($\hat{\sigma}^2$).

For testing procedures, this research first tested the hypothesis that there is no threshold effect ($H_0 : \beta_1 = \beta_2$), using the likelihood ratio: $F_1 = (S_0 - S_1(\hat{\gamma}))/\hat{\sigma}^2$, where $S_0$ and $S_1(\hat{\gamma})$ are the sum of squared error under hypothesis $H_0$ and the opposite hypothesis ($H_1 : \beta_1 \neq \beta_2$), respectively. However, since the asymptotic distribution of F1 is not normal, we used the bootstrap procedure to determine critical values and probability values (*p*-value). When the *p*-value is less than the desired condition value, we reject the $H_0$ hypothesis.

When a threshold effect exists ($\beta_1 \neq \beta_2$), Hansen (1999) considered that $\gamma$ is consistent with $\gamma_0$ (actual value of $\gamma$) and asymptotic distribution is not normal at a significant level. Therefore, we needed to check the asymptotic distribution of the estimated threshold with the hypothesis $H_0 : \gamma = \gamma_0$, by applying the likelihood ratio: $LR_1 = (S_1(\gamma) - S_1(\hat{\gamma}))/\hat{\sigma}^2$ with asymptotic confidence intervals of $c(\alpha) = -2 \log(1 - \sqrt{1 - \alpha})$, where $\alpha$ is the significance level (1%, 5%, and 10%). With the significance level $\alpha$ and $LR_1(\gamma_0) > C(\alpha)$, we can reject the hypothesis $H_0 : \gamma = \gamma_0$, meaning that the actual threshold value is not equal to the estimated threshold value.

If there exists a double threshold, the model can be modified as follows:

$$ROA_{i,t} = \begin{cases} \mu_i + \theta'H_{i,t} + \beta_1 CASH_{i,t} + \varepsilon_{it}, & \text{if } CASH_{i,t} \leq \gamma_1 \\ \mu_i + \theta'H_{i,t} + \beta_2 CASH_{i,t} + \varepsilon_{it}, & \text{if } \gamma_1 < CASH_{i,t} \leq \gamma_2, \\ \mu_i + \theta'H_{i,t} + \beta_3 CASH_{i,t} + \varepsilon_{it}, & \text{if } CASH_{i,t} > \gamma_2 \end{cases} \qquad (7)$$

where $\gamma_1 < \gamma_2$. This can be extended to multi-threshold models ($\gamma_1, \gamma_2, \gamma_3, \ldots, \gamma_n$).

According to Li (2016), in econometrics, the endogeneity problem arises when the explanatory variables and the error term are correlated in a regression model, leading to biased and inconsistent parameter estimates. Particularly, this problem plagues almost every aspect of empirical corporate finance. To solve for the endogeneity problem, among all the remedies, the generalized method of moments (GMM) has the greatest correction effect on the coefficient, followed by instrumental variables, fixed-effects models, lagged dependent variables, and control variables.

Earlier literature on corporate cash holdings showed that there exist problems of endogeneity and omitted variable bias (Ozkan and Ozkan 2004). The endogeneity problem might arise in cash literature for several reasons. For example, firm-specific characteristics are not strictly exogenous, and have shocks affecting firm performance as well as influencing dependent variable CASH like size and leverage. Additionally, the presence of dependent variables may be correlated with past and current residual terms. To solve for the endogeneity problem that appears in the empirical analysis of cash holdings and firm value, Martínez-Sola et al. (2013), Azmat (2014) and Nguyen et al. (2016) applied the dynamic regression model—GMM estimation.

The threshold regression methods by Hansen (1999) were developed for non-dynamic panels with individual specific fixed effects. Least squares estimation of the threshold and regression slopes was proposed using fixed-effects transformations. This method has the disadvantage that the independent variables in the model are exogenous assumptions, which may in fact be endogenous. Therefore,

this method explicitly excludes the presence of endogenous variables, and this has been an impediment to empirical application, including dynamic panel models.

## 4. Results and Discussions

### 4.1. Descriptive Statistics for Variables in the Model

Table 1 below presents descriptive statistic for the variables in the model. All of these variables were calculated based on the financial information collected from the balance sheet and income statement of 306 non-financial companies listed on the Vietnam stock exchange market in the period of 2008–2017.

**Table 1.** Descriptive statistic of variables.

| Variables | Observations | Mean | Median | SD | Min | Max |
|---|---|---|---|---|---|---|
| ROA | 3060 | 0.0955 | 0.0850 | 0.0991 | −1.6451 | 1.1362 |
| CASH | 3060 | 0.1049 | 0.0625 | 0.1189 | 0.0001 | 0.9546 |
| SIZE | 3060 | 26.6859 | 26.6279 | 1.4318 | 22.6384 | 31.6017 |
| MB | 3060 | 1.1012 | 0.8640 | 0.8965 | 0.1001 | 9.2005 |
| LEV | 3060 | 0.4852 | 0.5043 | 0.2306 | 0.0056 | 0.9982 |

Note: ROA represents company performance, measured by profit before tax and interest on total assets; CASH represents the percentage of cash held by the company, measured by the ratio of money and cash equivalents to total assets; MB represents company growth, measured by market value over book value of stocks; SIZE represents company size, measured by Ln (total assets); LEV represents company leverage, measured by total debt over total assets.

The statistical results described in Table 1 showed that the average ROA is 9.55%, indicating that in average with 1 VND (VND commonly refers to Vietnamese đồng, the currency of Vietnam) of capital used annually, firms can generate about 0.0955 VND profit before tax and interest. The average cash holding ratio (CASH) is 10.49%, indicating that cash and cash equivalents account for 10.49% of the company's total assets. The average firm's size is 26.6859, equivalent to 389 billion VND, the ratio of market value to book value is 1.1012 on average, and the average leverage (LEV) is 48.52%. Observation numbers, median values, standard deviations, and minimum and maximum values of variables are also presented in Table 1.

### 4.2. Stationary Test Results of Variables in the Model

In fact, the threshold regression model of Hansen (1999) is an extension of the ordinary least squares (denoted by OLS) traditional estimation method. This method requires that all variables considered in the model must be stationary variables to avoid spurious regression. This study uses the Levin et al. (2002) and Im et al. (2003) standards to test the stationarity of variables in the model. By using STATA software with the dataset described in Section 4.1, the results of the unit root test and stationarity test of the variables are shown in Table 2 below.

**Table 2.** Unit root test results.

| Variables | LLC | | IPS | |
|---|---|---|---|---|
| | *t*-Statistic | *p*-Value | *z*-Statistic | *p*-Value |
| ROA | −8.6771 | *** 0.0000 | −3.4253 | *** 0.0000 |
| CASH | −9.4776 | *** 0.0000 | −5.3907 | *** 0.0000 |
| SIZE | −7.6484 | *** 0.0000 | −2.2162 | ** 0.0133 |
| MB | −17.2837 | *** 0.0000 | −3.5020 | *** 0.0000 |
| LEV | −13.0545 | *** 0.0000 | −2.9177 | *** 0.0018 |

Note: LLC and IPS are unit root tests of Levin et al. (2002) and Im et al. (2003) respectively. *** and ** give 1% and 5% significance, respectively.

Table 2 shows that according to LLC and IPS accreditation standards, all variables representing profitability (ROA), cash holding (CASH), scale (SIZE), growth (MB), and leverage (LEV) are stationary sequence and statistically significant at 1% and 5%. Thus, the use of these variables in the threshold regression model is completely acceptable.

*4.3. Threshold Regression Results*

This study used GAUSS software and applied the bootstrap method to obtain an approximation of F-statistics and then calculated *p*-value. *F*-statistics include F1, F2 and F3, to evaluate $H_0$ hypotheses for zero, one, and two thresholds, respectively. Table 3 provides results of single-threshold, double-threshold and triple-threshold tests.

**Table 3.** Test results of threshold effect of cash holding ratio on firm's performance.

| Threshold Value | *F*-Statistics | | Test Critical Values | | |
|---|---|---|---|---|---|
| | *F*-Statistic | *p*-Value | 1% | 5% | 10% |
| Single-threshold test | | | | | |
| 0.0993 | 21.3377 | *** 0.004 | 18.1411 | 14.3497 | 12.8103 |
| Double-threshold test | | | | | |
| 0.0993 | | | | | |
| 0.1715 | 10.1255 | 0.156 | 17.4561 | 13.5491 | 17.4561 |
| Triple-threshold test | | | | | |
| 0.0722 | 7.4726 | 0.334 | 18.1409 | 13.1731 | 10.6866 |
| 0.0993 | | | | | |
| 0.1715 | | | | | |

Note: *F*-statistics and *p*-value were obtained by executing a repeating bootstrap procedure 500 times for each bootstrap test. *** indicates significance at 1%.

First of all, this study examined the existence of a single-threshold effect. By using bootstrap to perform 500 times, the obtained F1-statistics and *p*-value are 21.3377 and 0.004 (<1%), respectively. This suggested that the null hypothesis is rejected at the 1% significance level. Next, this study examined the existence of a double-threshold effect. Similarly, using bootstrap to perform 500 times, the obtained F2-statistics and *p*-value are 10.1255 and 0.156 (>10%), respectively. This suggested that the hypothesis that a double threshold is rejected. Finally, this study examined the existence of a triple-threshold effect. Similarly, by using bootstrap to perform 500 times, F3-statistics are 7.4726 and the *p*-value is 0.334 (>10%). This showed that the triple-threshold hypothesis is rejected.

Thus, the results of the threshold effect test showed that there is a single-threshold effect on cash holding and company efficiency. Figure 1 below shows the construction of confidence intervals for a single-threshold model.

Table 3 above presents the estimated values of the single threshold at 0.0993. The first-step threshold estimate is the point where the $LR_1(\gamma)$ equals zero, which occurs at $\hat{\gamma}_1 = 0.0993$. All observations in the sample were divided into two sets by the CASH threshold variable (above and below the threshold value of $\gamma = 0.0993$). Accordingly, this study identified two modes formed by threshold values from 0 to 9.93% and above 9.93%.

Table 4 shows the estimated coefficients, standard deviations according to the OLS, and White Methods for two models mentioned above. When the cash holding ratio (CASH) is smaller than 9.93%, the estimated coefficient $\hat{\beta}_1$ is 0.4078 and statistically significant at 1%, indicating that ROA will increase by 0.4078% when the cash holding ratio increases by 1%. When CASH is higher than 9.93%, the estimated coefficient $\hat{\beta}_2$ is 0.1556 and statistically significant at 1%, indicating that ROA will increase by 0.1556% when CASH increases by 1%. The results showed that the ROA regression coefficient by CASH is not a fixed value but depends on each threshold of cash holding ratio. Thus, it is clear that the relationship between cash holding ratio and operational efficiency (slope values)

varies according to different changes in cash holding ratio. This suggested the existence of a nonlinear relationship between cash holding ratio and company's performance.

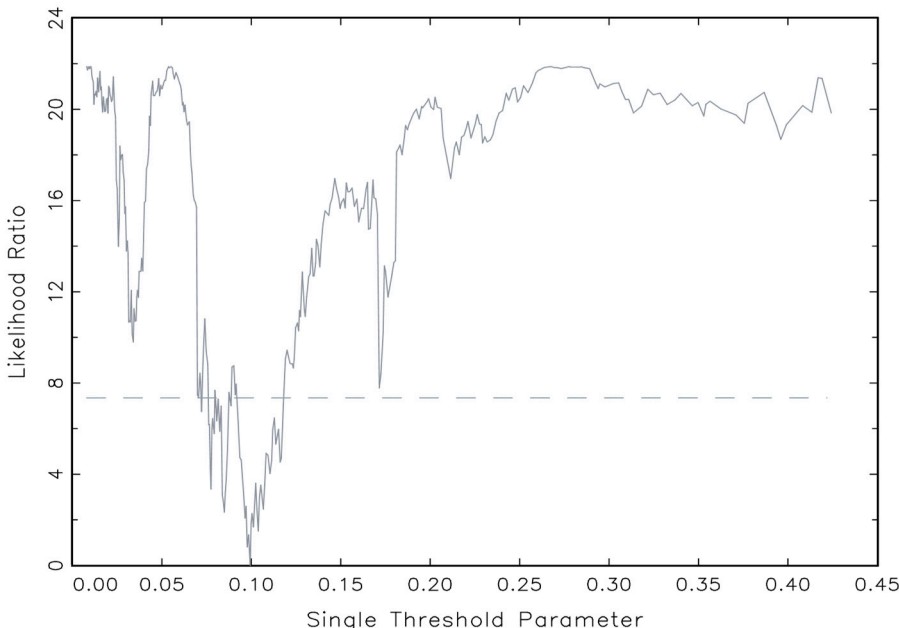

**Figure 1.** Confidence interval for the single-threshold model.

**Table 4.** Estimated results of regression coefficient for the cash holding ratio.

| Regression Coefficient | Estimated Value | OLS SE | $t_{\text{OLS}}$ | White SE | $t_{\text{White}}$ |
|:---:|:---:|:---:|:---:|:---:|:---:|
| $\hat{\beta}_1$ | 0.4078 | 0.0670 | *** 6.2620 | 0.0801 | *** 5.4124 |
| $\hat{\beta}_2$ | 0.1556 | 0.0187 | *** 8.8011 | 0.0340 | *** 5.1182 |

Note: $\hat{\beta}_1$ and $\hat{\beta}_2$ are the coefficients of the cash holding ratio variable corresponding to each value of the threshold. *** indicate the meaning of 1% respectively.

Table 5 shows the estimated coefficients, the standard deviation according to the OLS, and White methods of three control variables, company size, growth, and leverage.

**Table 5.** Estimated results of coefficients for control variables.

| Regression Coefficient | Estimated Value | OLS SE | $t_{\text{OLS}}$ | White SE | $t_{\text{White}}$ |
|:---:|:---:|:---:|:---:|:---:|:---:|
| $\hat{\theta}_1$ | 0.0135 | 0.0021 | *** 5.7517 | 0.0032 | *** 3.7767 |
| $\hat{\theta}_2$ | −0.0169 | 0.0042 | *** −5.8027 | 0.0057 | *** −4.9657 |
| $\hat{\theta}_3$ | −0.1146 | 0.0153 | *** −7.4835 | 0.0199 | *** −5.7470 |

Note: $\hat{\theta}_1$, $\hat{\theta}_2$, and $\hat{\theta}_3$ are the estimated coefficients of company's growth (MB), company's size (SIZE), and leverage (LEV). *** indicates significance at 1%.

Table 5 above shows the estimated coefficients, the standard deviation according to the OLS, and White methods of three control variables, company size, growth, and leverage.

Table 5 shows that the estimated coefficient of company's growth $\hat{\theta}_1$ is 0.0135 and has a positive relationship with ROA at the 1% level, implying that company's growth is a motivation to increase company efficiency. This result is consistent with empirical research by Abor (2005). Meanwhile, the estimated coefficient of company's size $\hat{\theta}_2$ is −0.0169 and is inversely related to ROA at the 1% level. This implied that scaling up the company is not an incentive to increase company efficiency. The empirical finding is consistent with Cheng et al. (2010), Martínez-Sola et al. (2013), and Nguyen

et al. (2016). At the same time, the estimated coefficient of company's leverage $\hat{\theta}_3$ is −0.1146 and is inversely related to ROA at the 1% level, suggesting that the use of more debt capital in the capital structure is harmful to firm's performance. This finding is consistent with the finding of Abor (2005), Nguyen et al. (2016), and Vijayakumaran and Atchyuthan (2017).

From Tables 4 and 5, the estimated model can be rewritten as follows:

$$\text{ROA}_{i,t} = \begin{cases} \mu_i + 0.0135MB_{i,t}-0.0169\text{SIZE}_{i,t}-0.1146\text{LEV}_{i,t} + 0.4078\text{CASH}_{i,t} + \varepsilon_{i,t} \text{ if CASH}_{i,t} \leq 9.93\% \\ \mu_i + 0.0135MB_{i,t}-0.0169\text{SIZE}_{i,t}-0.1146\text{LEV}_{i,t} + 0.1556\text{CASH}_{i,t} + \varepsilon_{i,t} \text{ if CASH}_{i,t} > 9.93\% \end{cases}.$$

Table 6 below shows the number of companies in each threshold by year.

**Table 6.** Number of companies in each threshold by year.

| Year | $\text{CASH}_{i,t}$ of ≤9.93% | | $\text{CASH}_{i,t}$ of >9.93% | |
|---|---|---|---|---|
| | Number | Percentage (%) | Number | Percentage (%) |
| 2008 | 212 | 69% | 94 | 31% |
| 2009 | 189 | 62% | 117 | 38% |
| 2010 | 197 | 64% | 109 | 36% |
| 2011 | 193 | 63% | 113 | 37% |
| 2012 | 195 | 64% | 111 | 36% |
| 2013 | 179 | 58% | 127 | 42% |
| 2014 | 184 | 60% | 122 | 40% |
| 2015 | 186 | 61% | 120 | 39% |
| 2016 | 202 | 66% | 104 | 34% |
| 2017 | 211 | 69% | 95 | 31% |
| **Total** | **1948** | **64%** | **1112** | **36%** |

Table 6 shows that about 64% of companies fall into the category of having a cash holding ratio within the threshold of 9.93% (meaning that about 179 to 212 companies fall into this threshold each year), and about 36% companies fall into the threshold of having a cash holding ratio above 9.93% (meaning that about 94–127 companies fall into this threshold each year).

## 5. Conclusions and Recommendations

The decision on the cash holding ratio could have a significant impact on firm's performance and value. This study used the threshold regression model of Hansen (1999) to examine the threshold effect of cash holding ratio on the performance of 306 listed non-financial companies in the Vietnam stock exchange market during the period of 2008–2017. ROA was used to represent company performance, and the ratio of money and cash equivalents on total assets (CASH) was used to represent the company's cash holding ratio.

Experimental results showed that the single-threshold effect exists between the ratio of cash holding and company's performance. In addition, the coefficient is positive when the cash holding ratio is less than 9.93%, which means a proportion of cash holding within this threshold could contribute to improvement of company's efficiency. The coefficient is positive but tends to decrease when the ratio of cash holdings is higher than 9.93%, implying that an increase in cash holdings ratio beyond this threshold will further reduce the company's performance. Therefore, this result might conclude that the relationship between cash holding ratio and firm's performance is a nonlinear relationship. These results are consistent with the trade-off theory, in that the optimal cash holding ratio is determined by a trade-off between marginal cost and profit margin of cash holdings (Opler et al. 1999). At the same time, this result is also consistent with some previous empirical research (Azmat 2014; Martínez-Sola et al. 2013; Nguyen et al. 2016). Among the control variables, firm size and leverage have a significant negative effect on company's performance whereas market-to-book value ratio of stocks has a significant positive effect on company's performance.

From the research results above, this study suggested a few recommendations for non-financial companies listed on the Vietnam stock exchange market in deciding the cash holding ratio as follows: Firstly, companies should not hold cash more than 9.93% of total assets. To ensure and improve the company's performance, the optimal range of cash holding ratio should be below 9.93%. Secondly, for companies that currently have a cash holding ratio higher than 9.93%, it is necessary to reduce the cash holding ratio to approach the optimal ratio as discussed above. In order to accomplish this task, it is necessary to identify the factors that affect the motive of holding cash, thereby having specific policies to adjust the cash holding ratio more suitable for each specific group of companies. From this idea, we will conduct research on the factors that affect the cash holding motive for each group of companies at each specific cash holding rate threshold. Hopefully, our next research results will provide practical suggestions in determining the optimal percentage of cash holdings to improve firm's performance and value.

This study has used panel threshold regression by Hansen (1999), that is, for non-dynamic panels, studies can be conducted by using extended threshold panels (for dynamic panels and considering the issue of endogeneity) and for more rigorous results. This would be a worthwhile subject for future research.

**Funding:** This research received no external funding.

**Acknowledgments:** We are grateful to the anonymous reviewers for their very helpful comments and suggestions.

**Conflicts of Interest:** The author declares no conflicts of interest.

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
