# Peer review of "Optimal Cash Holding Ratio for Non-Financial Firms in Vietnam Stock Exchange Market"

_jrfm, doi:10.3390/jrfm12020104_

Reviewer 1 Report

Given the importance of optimal cash holding on firm performance, the goal of this paper is to find out whether there exists a threshold ratio for non-financial firms in Vietnam. The results, indeed, show that there exists one single optimal cash holding which is about 9.93 percent. In other words, firms who hold cash below the threshold level should benefit by holding more cash, while this effect diminishes beyond the 9.93 checkpoint.

This paper begins the research purposes by discussing both the pecking order and the trade-off theory. The subsequence literature review generally falls into the Vietnam market, it is suggested to combine findings from other stock markets to distinguish why it is appropriate to carry out this model under this particular setting. Moreover, a subsection to discuss the institutional background in Vietnam is also recommended.

Regarding the Data and Methodology, it is suggested to deliver your hypothesis together with the single-threshold regression model clearly to readers. In other words, a separate subsection is needed to discuss the application of each variable together with the appropriate literature. Moreover, the regression model is different from Hansen (1999). For example, the ratio of long-term debt to assets is missing in the equation (2). Please also justify (e.g. reference support) why you select company size (SIZEi,t) and company growth (MBi,t) as control variables, instead of others. Is there any study using this model for other markets, e.g the U.S. market?

Likewise, if you are using a different regression model, then a comprehensive discussion is recommended to deliver a clear message to readers who might not be familiar with this area. Furthermore, it is suggested to include a paragraph, either in a footnote or in the appendix to discuss how you reach the results of your regression model (i.e., Hansen (1999) has included a detailed appendix section to derive the regression models).

Author Response

Thank you for your comments and suggestions. I edited the manuscript according to your comments and suggestions.

I hope you will accept my manuscript.

I look forward to hearing from you soon.

1.     Combine findings from other stock markets to distinguish why this model should be implemented in the context of Vietnam: I explained in the introduction (lines 33-47) 

·       Dittmar and Mahrt-Smith (2007) state that in 2003, the sum of all cash and cash equivalents represented more than 13% of the sum of all assets for large US firms. Al‐Najjar and Belghitar (2011) find that cash represents, on average, 9% of the total assets for UK firms. In Vietnam context, the cash and cash equivalents on total assets of firms accounts for more than 10%. Thus, cash represents a sizeable asset for firms. Cash management may therefore be a key issue for corporate financial policy.

·       According to Horioka and Terada-Hagiwara (2013), Asian firms are heavily constrained by borrowing limits and will hold more cash for future investments than firms in developed countries. Hence, our focus on an emerging country, Vietnam, allows us to offer a number of new insights beyond the existing studies of the relationship between cash holdings and firm’s performance. In Vietnam context, given the great opportunities and challenges now, companies need to focus on cash management, as lifeblood of the company. Therefore, the current question for listed companies on Vietnam stock exchange market is how to manage cash in order to improve operational efficiency, contribute to increase the company’s value. To solve this problem, listed companies on Vietnam stock exchange market need to know how cash holding ratio affects the firm’s performance.

2.     Regarding the Data and Methodology, ….

·       The problems you require additionally, I explained in the description section of the variables (lines 168-226).

·       Line 144-154: Martínez-Sola, García-Teruel, and Martínez-Solano (2013) used US industry’s data from 2001-2007, and found a nonlinear relationship between cash holding ratio and company’s value. They explain that the concave relationship between cash holdings and firm value exists because firms balance the costs and benefits of cash holdings to identify the optimal level of cash. Following this optimal level, firms will adjust their cash reserve to maximize firm value. This result was also discovered earlier by Shinada (2012). In Vietnam, Nguyen et al. (2016) investigates the nonlinear relationship between firm value and corporate cash holdings in a sample of non-financial Vietnamese firms from 2008 to 2013. Authors focus on both static and dynamic regressions to test for a nonlinear relationship. Their results reveal an ‘inverse U-shape’ relationship between firm value and cash holdings, which is in line with trade-off theory.

To the best of author’s knowledge, until now there has been no published research on the application of threshold regression model to study this relationship. Hence, by using the threshold regression model Hansen (1999), this study further fill the gap in the literature on the behavior of firms, and focus on evidence of cash policies and firm’s performance.

3.     I think the research results have been presented clearly. Therefore, the appendix is not necessary.

Best regards,

Nguyen Thanh Cuong

Reviewer 2 Report

The paper, though studying an interesting research area, is preliminary, which is reflected by the reference list of only 14 papers, super short introduction, only 9 pages of the whole paper, many typos and broken structure, etc.

Let me start with the big picture. The paper is suggesting a “one size fits all” strategy for cash holdings. Namely, firm or manager characteristics do not matter to cash holding; there is an optimal ratio for ANY firm and ANY management. This is hard to believe, given all firms are different and some firms need to hold LOTS of cash (e.g., Apple company), while some do not need any cash. In addition, cash holding is determined not only by the firm but also by the managers (e.g., Coles J. and Li F. 2019. Managerial Attributes, Incentives, and Performance. Review of Corporate Finance Studies forthcoming.). Furthermore, it can be based on industry peers or industry tournament effects (Coles, J. et al. 2018. Industry Tournament Incentives. Review of Financial Studies 31(4):1418-1459). The paper does not consider all these aspects. Discuss all these considerations and convince people where there exists an optimal ratio for all firms.

The paper seems to claim causality but doesn’t discuss the potential endogeneity issue and its remedies sufficiently. For example, if firm performs well, it receives lots of cash, a reverse causality. Or all of them are determined by firm fundamentals such as size, industry, performance, governance. See Li 2016, Endogeneity in CEO power: A survey and experiment, Investment Analysts Journal, 45 (3): 149-162 for a summary of methods to deal with the endogeneity problem. No need to use all these methods but at least discuss them in your scenario.

Related to the above point, you should study and rationalize the use of firm size measure, as a scale for cash holding ratio, in the literature since frim size is the key variable in this area and they affect all the variables you study simultaneously. See Dang et al. 2018. Measuring Firm Size in Empirical Corporate Finance. Journal of Banking & Finance, 86:159-176. After all it is the most significant variable in most studies alike. You need to discuss and justify your firm size measure.

The intro is too short. You should list your contributions up front in the intro, right after the motivation and findings. The conclusion should summarize all findings, implications, contributions and point out future research direction. You need to proofread the paper and extend and update your references.

There are so many typos and grammatical errors. For instance,

on line 67, share it with (not to)

line 72, those investments’ returns (not those investment’s returns.)

line 106, it is not a complete sentence

Hope these comments and suggestions can help further this study.

Author Response

Thank you for your comments and suggestions. I edited the manuscript according to your comments and suggestions.

I hope you will accept my manuscript.

I look forward to hearing from you soon.

1.     On the issue of exists the optimal cash holding rate for all firms: I explained in the introduction (lines 29-38) and the literature reviews (line 144-154).

·       Lines 29-38: Dittmar and Mahrt-Smith (2007) state that in 2003, the sum of all cash and cash equivalents represented more than 13% of the sum of all assets for large US firms. Al‐Najjar and Belghitar (2011) find that cash represents, on average, 9% of the total assets for UK firms. In Vietnam context, the cash and cash equivalents on total assets of firms accounts for more than 10%.

·       Line 144-154: Martínez-Sola, García-Teruel, and Martínez-Solano (2013) used US industry’s data from 2001-2007, and found a nonlinear relationship between cash holding ratio and company’s value. They explain that the concave relationship between cash holdings and firm value exists because firms balance the costs and benefits of cash holdings to identify the optimal level of cash. Following this optimal level, firms will adjust their cash reserve to maximize firm value. This result was also discovered earlier by Shinada (2012). In Vietnam, Nguyen et al. (2016) investigates the nonlinear relationship between firm value and corporate cash holdings in a sample of non-financial Vietnamese firms from 2008 to 2013. Authors focus on both static and dynamic regressions to test for a nonlinear relationship. Their results reveal an ‘inverse U-shape’ relationship between firm value and cash holdings, which is in line with trade-off theory.

To the best of author’s knowledge, until now there has been no published research on the application of threshold regression model to study this relationship. Hence, by using the threshold regression model Hansen (1999), this study further fill the gap in the literature on the behavior of firms, and focus on evidence of cash policies and firm’s performance.

2.     About discuss the potential endogeneity issue and its remedies sufficiently: This problem I explained in the article (line 268-270; line ) .

·       Line 268-270: The threshold regression method by Hansen (1999) has the disadvantage that the independent variables in the model are assumptions exogenous, which may in fact be endogenous. At the same time, Hansen (1999) panel threshold specifically designed for a balanced panel dataset and non-dynamic panels.

·       Line 397-399: This study has used panel threshold regression by Hansen (1999) that is for non-dynamic panels, studies can be conducted by using extended threshold panels (for dynamic panels and considering the issue of endogeneity) and for more rigorous results.

3.     The problem of measuring company size:  I also discussed in the article (line 196-205).

·       Firm size (SIZE) is considered one determinant of firm performance and value. Abor (2005), Vijayakumaran and Atchyuthan (2017) suggest that enterprises of higher size generally have higher firm performance. On the other hand, researches by Cheng, Liu, and Chien (2010), Martínez-Sola et al. (2013), Nguyen et al. (2016) suggest that enterprises of higher size generally have lower firm performance and value. Thus, the relationship between the size and the performance of companies is unclear.

·       To measure firm size, there exist different perspectives. According to Nguyen et al. (2016), Vijayakumaran and Atchyuthan (2017), firm size is defined by natural logarithm of total assets. Further, Martínez-Sola et al. (2013) show that firm size is defined by natural logarithm gross sales. In this study, we only use book value of total asset to calculate firm size.

4.     I listed the contributions up front in the introduction, right after motivation and discovery. The conclusion also summarizes all the findings, implications, contributions and points out future research direction. I also edited spelling and grammar errors in the article.

Best regards

Nguyen Thanh Cuong

Round  2

Reviewer 1 Report

I am fine with the revision, and decided to accept this paper.

Author Response

Thank you for your acceptance of my article.

I also edited spelling and grammar errors in the article. 

Kind regards,

Nguyen Thanh Cuong

Reviewer 2 Report

Seems you did not address my concerns sufficiently. I have a lot more comments, not included in your response file. Please take time to address each in detail.

Author Response

Thanks for your comments and suggestions. 

I send you my responses on your comments and suggestions (see attached). I look forward to your acceptance.  

I look forward to hearing from you soon.

Kind regards,

Nguyen Thanh Cuong

Round  3

Reviewer 2 Report

Improved